Comparing sediment DNA extraction methods for assessing organic enrichment associated with marine aquaculture

Pearman John K. 1 John.Pearman@cawthron.org.nz
http://orcid.org/0000-0002-3397-8457 Keeley Nigel B. 2
http://orcid.org/0000-0003-1976-8266 Wood Susanna A. 1
http://orcid.org/0000-0003-0755-0083 Laroche Olivier 2
Zaiko Anastasija 1 3
http://orcid.org/0000-0001-8337-5489 Thomson-Laing Georgia 1
http://orcid.org/0000-0001-8749-1495 Biessy Laura 1
http://orcid.org/0000-0003-0625-3431 Atalah Javier 1
Pochon Xavier 1 3
1 Coastal and Freshwater Group, Cawthron Institute , Nelson , New Zealand
2 Institute of Marine Research in Norway , Tromsø , Norway
3 Institute of Marine Science, University of Auckland , Auckland , New Zealand
Wangensteen Owen
Electronic publication date: 2020 Oct 27
Publication date: 2020
Volume: 8
Electronic Location ID: e10231
Received 2020 Aug 17; Accepted 2020 Oct 2
Copyright: © 2020 Pearman et al.
Copyright year: 2020
Copyright holder: Pearman et al.
License: This is an open access article distributed under the terms of the Creative Commons Attribution License, which permits unrestricted use, distribution, reproduction and adaptation in any medium and for any purpose provided that it is properly attributed. For attribution, the original author(s), title, publication source (PeerJ) and either DOI or URL of the article must be cited.
License URL: https://creativecommons.org/licenses/by/4.0/

Keywords: Metabarcoding, DNA extraction, 16S rRNA gene, 18S rRNA gene, Environmental DNA, Benthic monitoring

Funding: New Zealand Seafood Innovation Limited 1804 New Zealand King Salmon Co. Limited Cawthron Institute Internal Investment Fund New Zealand Ministry for Primary Industry Marlborough District Council This research was supported by multiple agencies, including (in reducing order of contribution): New Zealand Seafood Innovation Limited (contract 1804), New Zealand King Salmon Co. Limited, Cawthron Institute Internal Investment Fund, the New Zealand Ministry for Primary Industry, and the Marlborough District Council. The funders had no role in study design, data collection and analysis, decision to publish, or preparation of the manuscript.

==============================
Marine sediments contain a high diversity of micro- and macro-organisms which are important in the functioning of biogeochemical cycles. Traditionally, anthropogenic perturbation has been investigated by identifying macro-organism responses along gradients. Environmental DNA (eDNA) analyses have recently been advocated as a rapid and cost-effective approach to measuring ecological impacts and efforts are underway to incorporate eDNA tools into monitoring. Before these methods can replace or complement existing methods, robustness and repeatability of each analytical step has to be demonstrated. One area that requires further investigation is the selection of sediment DNA extraction method. Environmental DNA sediment samples were obtained along a disturbance gradient adjacent to a Chinook (Oncorhynchus tshawytscha) salmon farm in Otanerau Bay, New Zealand. DNA was extracted using four extraction kits (Qiagen DNeasy PowerSoil, Qiagen DNeasy PowerSoil Pro, Qiagen RNeasy PowerSoil Total RNA/DNA extraction/elution and Favorgen FavorPrep Soil DNA Isolation Midi Kit) and three sediment volumes (0.25, 2, and 5 g). Prokaryotic and eukaryotic communities were amplified using primers targeting the 16S and 18S ribosomal RNA genes, respectively, and were sequenced on an Illumina MiSeq. Diversity and community composition estimates were obtained from each extraction kit, as well as their relative performance in established metabarcoding biotic indices. Differences were observed in the quality and quantity of the extracted DNA amongst kits with the two Qiagen DNeasy PowerSoil kits performing best. Significant differences were observed in both prokaryotes and eukaryotes (p < 0.001) richness among kits. A small proportion of amplicon sequence variants (ASVs) were shared amongst the kits (~3%) although these shared ASVs accounted for the majority of sequence reads (prokaryotes: 59.9%, eukaryotes: 67.2%). Differences were observed in the richness and relative abundance of taxonomic classes revealed with each kit. Multivariate analysis showed that there was a significant interaction between “distance” from the farm and “kit” in explaining the composition of the communities, with the distance from the farm being a stronger determinant of community composition. Comparison of the kits against the bacterial and eukaryotic metabarcoding biotic index suggested that all kits showed similar patterns along the environmental gradient. Overall, we advocate for the use of Qiagen DNeasy PowerSoil kits for use when characterizing prokaryotic and eukaryotic eDNA from marine farm sediments. We base this conclusion on the higher DNA quality values and richness achieved with these kits compared to the other kits/amounts investigated in this study. The additional advantage of the PowerSoil Kits is that DNA extractions can be performed using an extractor robot, offering additional standardization and reproducibility of results.

Introduction

Marine sediments harbor diverse biological communities that are vital in maintaining biogeochemical cycles, food webs and ecosystem functioning. However, these communities can be significantly impacted by anthropogenic activities (Snelgrove, 1997; Dell’Anno et al., 2003). Studies assessing the effects of human induced perturbations on the benthic environment have traditionally involved the analysis of communities of macro-organisms (Papageorgiou, Sigala & Karakassis, 2009; Keeley, Forrest & Macleod, 2013; Aguado-Giménez et al., 2015). Micro- and meio-benthic organisms have received less attention partly due to the challenges associated with morphologically identifying the immense diversity of these organisms. Developments in environmental genomics now allow communities to be more accurately characterized. These techniques are currently being touted as cost-effective and sensitive methodologies to monitor entire biological communities in marine sediments, especially along gradients of anthropogenic disturbances (Pawlowski et al., 2016b; Laroche et al., 2016, 2018; Aylagas et al., 2017; Borja, 2018; Keeley, Wood & Pochon, 2018; Cordier, 2020).

Environmental genomic techniques, which enable a broad range of taxonomic groups to be characterized from environmental DNA (eDNA), have become more prevalent in the last decade (Taberlet et al., 2018). The DNA in these samples originates from a combination of microbes, organisms body parts or cells contained in feces, epidermal mucus, urine, saliva and gametes of larger organisms (Rees et al., 2014; Taberlet et al., 2018). While eDNA-based techniques (e.g., metabarcoding) are now used extensively in ecological studies (Bohmann et al., 2014), they are also considered for routine biomonitoring purposes (Aylagas et al., 2020). The delay in their incorporation into monitoring regimes is due, in part, to the need for each step of the process (e.g., sediment collection, DNA extraction, PCR amplification, etc.) to be demonstrated as robust and repeatable (Darling et al., 2020).

Environmental DNA methodologies have the potential for monitoring of a variety of disturbance gradients but an eDNA application that is close to uptake and implementation is the use of metabarcoding for monitoring benthic impacts of fish farming (Aylagas et al., 2020). Sea-cage-based fish farms are inevitably associated with elevated fluxes of organic waste, often culminating in severe localized benthic enrichment (i.e., anoxic and azoic conditions directly beneath the farms), which gradually decreases with distance from the fish cages (Brooks & Mahnken, 2003a, 2003b). Routine monitoring of the benthic environment is usually required by regulation and traditional monitoring methods typically involve measuring the chemical properties of sediment and microscopic analysis of macrofaunal diversity (Keeley, Macleod & Forrest, 2012). In New Zealand for example, these parameters are incorporated into an Enrichment Stage (ES) index (Keeley, Macleod & Forrest, 2012; Keeley et al., 2012; MPI, 2018), which provides regulators and producers with an integrated, weight-of-evidence-based measure of environmental impact. Morphological approaches, however, are often time-consuming, expensive, and require a high level of taxonomic expertise that is shrinking globally (Jones, 2008). These limitations have led to numerous metabarcoding investigations describing the ecological responses of a wide range of organisms associated with enrichment states, including bacteria (Fodelianakis et al., 2015; Dowle et al., 2015; Verhoeven et al., 2018; Stoeck et al., 2018a), foraminifera (He et al., 2010; Pawlowski et al., 2014, 2016a; Pochon et al., 2015), ciliates (Stoeck et al., 2018b), metazoans (Lejzerowicz et al., 2015), or a combination of multi-trophic taxa (Keeley, Wood & Pochon, 2018; Frühe et al., 2020). Although all of these studies have revealed consistent organismal responses to fish farm enrichment, indicating that metabarcoding is a cost-effective tool for routine monitoring, they have all used different sediment collection methods, varying amounts of starting material (from 0.25 g to 10 g of sediment), and a variety of DNA extraction kits. The succesful uptake of metabarcoding tools for commercial monitoring of fish farms requires a fully standardized and validated laboratory workflow. There is a need to evaluate the effects that DNA extraction methods have on the detection of bioindicator taxa and the entire community diversity.

One of the complexities with analysis of sediment samples is that they are remarkably variable in chemical composition and physical properties across spatial scales. This variability can have an impact on the absorption of eDNA, with clays and humic acids having a strong binding capacity for DNA molecules (Dell’Anno, Stefano & Danovaro, 2002). Other factors such as temperature and porewater pH have an impact on the retention and stability of DNA (Levy-Booth et al., 2007; Torti, Lever & Jørgensen, 2015). This has led to a multitude of specific protocols that aim to optimize the extraction process for different sediment types (Kallmeyer & Smith, 2009; Morono et al., 2014; Lever et al., 2015). These methods rely on disrupting the cell membranes by either physical (e.g., bead-beating, freeze-thaw; MacGregor et al., 1997; Haile, 2012; Pearman et al., 2016), chemical (e.g., solvents; Pitcher, Saunders & Owen, 1989) or enzymatic (Holben et al., 1988) means. More recently, commercial DNA extraction kits have been used as an alternative to manual extraction protocols (Lear et al., 2018). There are drawbacks to commercial kits in that the protocols are often inflexible and the reagents are proprietary, but they streamline the often-laborious task of manual extracts and tend to avoid the use of organic solvents (Lever et al., 2015).

Previous research has shown that the quantity and quality of the extracted DNA can vary between methods (Knauth, Schmidt & Tippkötter, 2013; Lekang, Thompson & Troedsson, 2015; Hermans, Buckley & Lear, 2018; Armbrecht et al., 2020). Extraction methods have also been shown to affect detected bacterial composition with various phyla being either over or under represented (Luna, Dell’Anno & Danovaro, 2006; Carrigg et al., 2007; Holmsgaard et al., 2011; Lekang, Thompson & Troedsson, 2015; Hermans, Buckley & Lear, 2018). This has important implications for molecular-based biomonitoring, as results need to be comparable spatially and temporally, and should be repeatable and provide a true representation of the community in a sample. The majority of benthic marine studies that have targeted both prokaryotic and eukaryotic organisms have typically used kits that necessitate >2 g of sediment for DNA extraction (Lear et al., 2018). If the same results can be obtained with smaller volumes of starting material, then this will allow the extraction process to be automated (e.g., using a sample prep robot), which would standardize and greatly expedite the process and make it more cost-effective for routine monitoring.

The sampling described in this study was part of a long term (8 years) research project, which aimed to validate a metabarcoding-based protocol for assessing and monitoring the benthic impacts of salmon farming in New Zealand (Pochon et al., 2020). While interesting ecological inferences can be gained from studying impact gradients around fish farms those results will be presented elsewhere. The current study focusses on the first step in the workflow optimization process by investigating the effect of DNA extraction kits and sediment quantities on prokaryotic and eukaryotic assemblages along an organic enrichment gradient adjacent to fish farms. The null hypothesis of the experiment was that distinct extraction kits and sediment quantities would not affect the quality or quantity of extracted DNA, detected prokaryotic and eukaryotic assemblages, and therefore—metabarcoding-based benthic health assessment.

Methods

Sediment eDNA samples were collected in November 2015 as part of a regular compliance monitoring program for New Zealand King Salmon (NZKS) at a Chinook (Oncorhynchus tshawytscha) salmon farm located in Otanerau Bay (41°10′11″S, 174°19′16″E), Marlborough Sounds, New Zealand (Fig. 1). The farm location is characterized as a “low flow” area, having a mean current velocity of approximately 6 cm s−1 and a water depth of 34–35 m. Sampling started directly alongside the pen and radiated outwards along an enrichment gradient with samples collected at 50 and 150 m, and at a control site located 625 m from the farm. Bulk sediment at each site was collected using a Van-Veen grab with sediment biogeochemical properties assessed at each station in Table S1. Three distinct surface-sediment samples (c. 40 g per grab) were collected from the top 1–2 cm of each grab using a sterile spatula and placed in DNA/RNAse-free collection tubes (50 mL). Sediment samples were immediately placed on ice and transported to the Cawthron laboratory where they were stored at −80 °C until DNA extraction.

Figure 1 Sampling sites around the Otanerau salmon farm in the Marlborough Sounds, New Zealand.

Site map indicating; (A) the location of the Otanerau salmon farm (OTA, red dot) within the Marlborough Sounds, New Zealand, and (B) the arrangement of the sampling stations with distance in relation to the OTA salmon farm. Figure modified from Dowle et al. (2015). Copyright © 2015, Oxford University Press.

Each step of the following molecular analyses was conducted in separate sterile laboratories dedicated to these steps, with sequential workflow to ensure no cross-contamination. Rooms dedicated to DNA extraction, amplification set-up and template addition were equipped with laminar flow cabinets with HEPA filtration and room-wide ultra-violet sterilization which was switched on for >15 min before and after each use. Aerosols barrier tips (Axygen, San Francisco, CA, USA) were used throughout.

Each of the 12 sediment samples were homogenized using sterile stainless steel grinding beads in the 1600 MiniG® tissue homogenizer (1,500 RPM, 2 min). From these homogenized samples five sub-samples representing three distinct sediment volumes (0.25, 2, and 5 g) were extracted using four DNA extraction kits (Table 1). Amongst the many kits available on market, we selected these four specific kits for the following reasons. Qiagen Power Soil kits (Q.PS and Q.PS.Pro) integrates a patented Inhibition Removal Technology® that works particularly well for eDNA isolation from challenging samples such as enriched soils. This likely explains why the majority of soil eDNA studies use Qiagen kits (Lear et al., 2018) and justifies our emphasis on Qiagen Power Soil kits in this study. Second, the Qiagen RNeasy PowerSoil Total RNA/DNA extraction/elution kit (QIA2) is the most commonly used kit in previous fish farm studies (Pawlowski et al., 2014; Dowle et al., 2015; Lejzerowicz et al., 2015; Pochon et al., 2015; Keeley, Wood & Pochon, 2018), but this DNA/RNA co-extraction protocol is comparatively very time-consuming and involves dangerous chemicals such as phenol-chloroform. Third, to our knowledge there are only two commercial soil kits that allow extraction of up to 10 g of material, the Qiagen PowerMax Soil kit and the Favorgen Soil Midi-prep kit (Young et al., 2014). Cost-considerations are essential for routine monitoring, and therefore we chose to test the latter kit, being significantly cheaper than the former kit. DNA was extracted using a QIAcube automated sample prep robot (Qiagen Instruments, Hombrechtikon, Switzerland) for the Qiagen DNeasy PowerSoil Kit (Q.PS) and Qiagen DNeasy PowerSoil Pro Kit (Q.PS.Pro) kits, while extraction was done manually for the remaining kits, according to the manufacturer’s instructions. The quality and purity of isolated DNA were measured using a Nanophotometer NP80 (Implen, Munich, Germany). This instrument is equipped with an automatic quality control software that enables the detection of impurities and/or air bubbles within extracted samples.

Table 1 Kits used in this study and sediment weights used with each kit.

Kit name	Abbreviated name	Sediment weight (g)	
Qiagen DNeasy PowerSoil Kit	Q.PS	0.25	
Qiagen DNeasy PowerSoil Pro Kit	Q.PS.Pro	0.25	
Qiagen RNeasy PowerSoil Total RNA/DNA extraction/elution Kit	QIA2	2	
Favorgen FavorPrep Soil DNA Isolation Midi Kit	FAV2	2	
Favorgen FavorPrep Soil DNA Isolation Midi Kit	FAV5	5	

Polymerase chain reactions (PCR) were performed on all extracted samples (n = 60) and targeted two genes. Prokaryotic communities were amplified using a 16S rRNA gene (V3-V4 region) with the primer set 341F: 5′-CCT ACG GGN GGC WGC AG-3′ (Herlemann et al., 2011) and 805R: 5′-GAC TAC HVG GGT ATC TAA TCC-3′ (Klindworth et al., 2013). Eukaryotic communities were targeted using the primer set Uni18SF: 5′-AGG GCA AKY CTG GTG CCA GC-3′ and Uni18SR: 5′-GRC GGT ATC TRA TCG YCT T-3′ (Zhan et al., 2013), which amplified the 18S rRNA gene (V4 region). Both the prokaryotic and eukaryotic primers had an Illumina overhang adapter present as per the Illumina 16S library preparation manual. Amplifications were undertaken in an Eppendorf Mastercycler (Eppendorf, Hamburg, Germany) in a total volume of 50 μL using MyFi™ PCR Master Mix (Bioline Meridian Bioscience, Memphis, Tennessee, USA), 2 μL of each primer (10 μM stock) and 2 μL of template eDNA. The PCR cycles for the 16S rRNA gene amplification were as follows: 95 °C for 5 min followed by 35 cycles of 94 °C (30 s), 54 °C (30 s) and 72 °C (45 s) with a final extension at 72 °C for 7 min. Amplifications for the 18S rRNA gene were 95 °C for 5 min followed by 37 cycles of: 94 °C (30 s), 54 °C (30 s) and 72 °C (45 s) with a final extension at 72 °C for 7 min. Negative PCR controls were included in each PCR run. Amplicon PCR products were purified using AMPure® XP PCR Purification beads (Agencourt®, Beverly, MA, USA) and quantified using a Qubit® Fluorometer (Life Technologies, Carlsbad, CA, USA). An additional water control was added to test for potential contamination during the following sequencing workflow. All negatives were subsequently sequenced. Products (n = 68; 60 samples plus 5 PCR blanks and 3 water blanks) were diluted (3 ng μL−1) and sent to Auckland Genomics (University of Auckland) for final library construction. Dual indices were added to the amplicons via a second round of PCR amplification as detailed in the Illumina 16S library preparation manual. Subsequent to the second round of amplification, 5 μL of each sample (including all controls) was pooled and a single clean-up was undertaken. A bioanalyzer was used to check the quality of the library which was then diluted 4 nM and denatured. The library was diluted to a final loading concentration of 7 ρM with a 15% spike of PhiX. Paired-end sequences (2 × 250 bp) were generated on an Illumina MiSeq instrument. Raw sequences were deposited in the NCBI short read archive under accession: PRJNA657189

Reads were demultiplexed using the MiSeq Reporter (v2) based on the Nextera™ dual-indexing. Primers were removed from the sequences using cutadapt (version 1.8); (Martin, 2011), allowing a maximum error rate of 0.1. Sample reads were processed using the DADA2 program (Callahan et al., 2016) implemented in QIIME2 version 2018.11 (Bolyen et al., 2019) using default parameters. The reads were truncated at 228 and 216 bp for the forward and reverse 16S rRNA gene sequences, and 225 and 216 bp for the 18S rRNA gene sequences, and the maxEE value (expected error rate) was set to 2. The sequences were merged into Amplicon Sequence Variants (ASVs) with a minimum overlap of 10 bp and no mismatches. Chimeras were detected and removed using the removeBimeraDenovo script in DADA2. Taxonomic assignments were undertaken in DADA2 based on the rdp (Wang et al., 2007) algorithm against the SILVA 132 database (Pruesse et al., 2007). In the prokaryotic dataset, ASVs assigned to eukaryotes, chloroplasts and mitochondria were removed prior to further analysis. Code for the analysis can be found at: https://github.com/olar785/Optimizing-DNA-extraction-methods-for-assessing-organic-enrichment-in-marine-farm-sediments/blob/master/Q2_DADA2_pipeline.sh and the taxonomy for each ASV sequence can be found in Table S2.

The output from DADA2 was imported into phyloseq (McMurdie & Holmes, 2013) within R software (R Core Team, 2020). To remove possible contamination from the data we used the maximum sequence count for each ASV present in the controls as a basis for subtraction (Bell et al., 2019). Thus, any ASV in the dataset with fewer reads than found in the controls was assumed to be contamination and removed from analysis. ASVs that had read numbers higher than the threshold had their read counts reduced by the threshold number to take into account the contamination. To allow comparison between samples, rarefaction plots were constructed with ggrare (Kandlikar et al., 2018) and ggplot2 (Wickham, 2016) and subsequently reads were subsampled to 4,400 per sample for prokaryotes and 10,000 per sample for eukaryotes (Fig. S1). Richness values were tested for normality (shapiro.test) and homogeneity of variance (bartlett.test) and subsequently a square root transformation was undertaken to meet these assumptions. Differences in richness (square root transformed) were assessed using two-way analysis of variance (ANOVA), with kit (5 levels) and distance (4 levels) as factors. Pairwise post-hoc tests were undertaken using the Tukey Honestly Significant Difference (HSD) test. Shared ASVs were assessed in phyloseq and plotted with VennDiagram (Chen & Boutros, 2011).

Multivariate analysis was undertaken on both datasets using the rarefied samples. Non-metric multidimensional scaling (nMDS) was undertaken to visualize the 2D representation of the community structure. Statistical differences were tested using permutational multivariate analysis of variance (PERMANOVA (Anderson, Gorley & Clarke, 2008)) based on Bray-Curtis dissimilarities of the square root transformed data using PRIMER (Anderson, Gorley & Clarke, 2008). The experimental design consisted of two crossed factors: Kit and Distance; five levels for factor Kit (Q.PS, Q.PS.Pro, QIA2, FAV2 and FAV5) and 4 levels for Distance (Pen, 50 m, 150 m and Control). To assess the taxonomic composition of the communities, ASVs were merged at class level. To assess the effect of kit on benthic health assessments the denovo indices, the bacterial Metabarcoding Biotic Index (b-MBI) and the eukaryotic Metabarcoding Biotic Index (e-MBI) were calculated using pre-defined molecular Eco-Groups at the ASV level following Keeley, Wood & Pochon (2018). Figures were constructed in R using the package ggplot2 (Wickham, 2016) and ampvis2 (Andersen et al., 2018).

Code for the statistical analysis can be found at: https://github.com/jkpearmanbioinf/FishFarmAnalysis/blob/master/KitComparison.notebook.Rmd.

Results

DNA quality was generally highest for samples extracted with the Q.PS kit (Table S3). The Q.PS.Pro kit yielded similar albeit more scattered absorbance values, and a lower overall DNA concentration compared to the Q.PS kit. Both the QIA2 and the FAV2 kits had low A260/A230 ratios indicating contamination by compounds that absorb in the A230 range (e.g., Ethylenediaminetetraacetic acid (EDTA), carbohydrates). The FAV5 and QIA2 kits yielded the highest overall DNA concentration estimates, although they failed most of the automatic quality controls (Table S3). Lower DNA concentrations after PCR cleanup were noted for the FAV2, FAV5 and QIA2 kits in the eukaryotic samples compared with the Q.PS and Q.PS.Pro kits (Table S3).

High-Throughput Sequencing resulted in a total of 3,337,510 prokaryotic sequences (915,508 after filtering; Table S4) and 6,318,916 eukaryotic sequences (4,382,737 after filtering; Table S4). Replicates for FAV2 at 150 m for the prokaryotic dataset were removed from the dataset as they did not meet the rarefaction thresholds. Following bioinformatics analyses using DADA2, a total of 14,427 and 11,177 ASVs were identified for prokaryotic and eukaryotic communities, respectively.

There was a statistical difference in the observed richness amongst kits for the eukaryotic (F = 7.442, p < 0.001) dataset, while there was a significant interaction in the prokaryotic dataset (F = 7.575; p < 0.001). Pairwise tests showed that there was has a higher diversity retrieved in the Q.PS and Q.PS.Pro kits in the pen compared with other kits in the prokaryotic dataset (Fig. 2A) with the majority of the other comparisons non-significant. The FAV2 kit had a significantly lower diversity than the other kits in the eukaryotic dataset (Fig. 2B). Similar trends were observed when investigating the Shannon diversity for the prokaryotes (F = 13.87, p < 0.001, Fig. S2). There was a significant trend for the eukaryotes (F = 2.723, p = 0.039, Fig. S2) although no pairwise comparisons were significant.

Figure 2 Number of amplicon sequence variants for different DNA extraction kits.

Number of Amplicon Sequence Variants per kit for the eukaryotic 18S rRNA (Eukaryotes; A) gene and prokaryotic 16S rRNA (Prokaryotes; B) gene. The symbols designate the distance from the aquaculture pen. Q.PS, Qiagen Dneasy PowerSoil Kit (0.25 g); Q.PS.Pro, Qiagen Dneasy PowerSoil Pro Kit (0.25 g); QIA2, Qiagen Rneasy PowerSoil Total RNA/DNA extraction/elution Kit (2 g); FAV2, Favorgen FavorPrep Soil DNA Isolation Midi Kit (2 g); FAV5, Favorgen FavorPrep Soil DNA Isolation Midi Kit (5 g).

Only a small proportion of the ASVs were shared amongst all kits (prokaryotes: 3.5% eukaryotes: 3.2%), however these shared ASVs accounted for 59.9% of prokaryotic and 67.1% of eukaryotic reads (Fig. 3). This indicates that the majority of the ASVs that are not shared are of low abundance. In the prokaryotic dataset, the Q.PS.Pro kit had the greatest proportion of unique (not observed in any other kit) ASVs accounting for 66.5% of the total prokaryotic diversity retrieved by the kit (Fig. 3A). The lowest proportion of unique ASVs was observed in the FAV5 (43.9%) and FAV2 (44.3%) kits for the prokaryotic dataset. For the eukaryotic dataset, all kits had a similar number of unique ASVs, ranging from 51% in the Q.PS.Pro kit to 56.2% in the QIA2 kit (Fig. 3B).

Figure 3 Shared amplicon sequence variants amongst kits.

Number of shared Amplicon Sequence Variants amongst kits for the; (A) prokaryotic 16S rRNA gene, and (B) eukaryotic 18S rRNA gene datasets. Q.PS, Qiagen Dneasy PowerSoil Kit (0.25 g); Q.PS.Pro, Qiagen Dneasy PowerSoil Pro Kit (0.25 g); QIA2, Qiagen Rneasy PowerSoil Total RNA/DNA extraction/elution Kit (2 g); FAV2, Favorgen FavorPrep Soil DNA Isolation Midi Kit (2 g); FAV5, Favorgen FavorPrep Soil DNA Isolation Midi Kit (5 g).

In general, all kits detected the same dominant classes of both prokaryotes and eukaryotes. However, the relative abundance of these groups differed when all distances from the farm were combined (Fig. 4; Fig. S3). The dominant prokaryotic class Bacteroidia yielded similar percentages of read abundances across all kits (Fig. 4A). Nonetheless, there were differences at the family level, with Flavobacteriaceae having lower relative abundances in the Q.PS (12.2%) and Q.PS.Pro (7.4%) kits compared with the other kits (FAV2: 24.7%; FAV5: 18.7% and QIA2: 22.1%) while Bacteroidetes BD2-2 had higher relative abundances in these kits (Q.PS: 9% and Q.PS.Pro: 8.3%) than the others (FAV2: 3.7%; FAV5: 3.4% and QIA2: 3.7%). Campylobacteria was the second most abundant class across kits except for the Q.PS and Q.PS.Pro kits (Fig. 4A). Fusobacteria and Deltaproteobacteria had higher relative abundances in the Q.PS and Q.PS.Pro kits compared with the other three kits. Differences in the relative abundance of reads in the eukaryotic dataset was more variable at the class level between kits (Fig. 4B). The apicomplexan class Conoidasida had a higher relative read abundance in the FAV2 and FAV5 kits and was especially low in the Q.PS kit. In contrast, Chromadorea (Nematoda), was lower in these kits (FAV2 and FAV5) compared with the other kits especially Q.PS and Q.PS.Pro (Fig. 4B). While ASVs belong to fish were not a substantial proportion of the dataset they were detected as would be expected with higher read numbers underneath the pens. Furthermore, there were differences in the number of ASVs found per class across kits in both the 16S rRNA and 18S rRNA gene datasets (Tables S5 and S6, respectively).

Figure 4 Relative abundance of sequence reads for the 10 most abundant classes.

Relative abundances (% of sample total) of sequence reads corresponding to the ten most abundant classes of; (A) prokaryotes, and (B) eukaryotes. A complete breakdown of all classes is presented in Tables S4 and S5. Q.PS, Qiagen Dneasy PowerSoil Kit (0.25 g); Q.PS.Pro, Qiagen Dneasy PowerSoil Pro Kit (0.25 g); QIA2, Qiagen Rneasy PowerSoil Total RNA/DNA extraction/elution Kit (2 g); FAV2, Favorgen FavorPrep Soil DNA Isolation Midi Kit (2 g); FAV5, Favorgen FavorPrep Soil DNA Isolation Midi Kit (5 g).

Similar patterns were observed for the community structure (Fig. 5) amongst kits with distance from the farm (i.e., decrease in organic enrichment levels) being a stronger determinant than kit. Strong and consistent clustering of replicate samples and kits were observed amongst collection sites, from highly enriched sediments adjacent to fish farm pens through to the un-enriched control sites, particularly in the prokaryotic dataset (Fig. 5A). Although similar patterns were observed for eukaryotes, the clusters appeared to be more diffused compared to prokaryotes (Fig. 5B). Both the prokaryote (F = 1.71; p < 0.001) and eukaryote (F = 1.36; p < 0.001) PERMANOVA results confirmed that there was significant interaction between kits and distance (Table S7). Pairwise comparisons indicated that at each distance there was no significant difference amongst kits (with 3 exceptions; see Table S7). For factor kit pairwise comparisons indicated that the Q.PS and Q.PS.Pro kits were better at differentiating the environmental gradient with pairwise comparisons significantly different amongst distances for these two kits (Table S7).

Figure 5 Non metric multi-dimensional scaling plots for the prokaryotic and eukaryotic community structures.

Non metric multi-dimensional scaling plots depicting; (A) prokaryotic (16S rRNA gene), and (B) eukaryote (18S rRNA gene) species community structures (Bray Curtis distance matrix of square root transformed relative abundance data) across a distance gradient at Otanerau fish farm in New Zealand (2015). The plot displays the clustering of communities recovered using the five distinct DNA extraction methods. Q.PS, Qiagen Dneasy PowerSoil Kit (0.25 g); Q.PS.Pro, Qiagen Dneasy PowerSoil Pro Kit (0.25 g); QIA2, Qiagen Rneasy PowerSoil Total RNA/DNA extraction/elution Kit (2 g); FAV2, Favorgen FavorPrep Soil DNA Isolation Midi Kit (2 g); FAV5, Favorgen FavorPrep Soil DNA Isolation Midi Kit (5 g). Colored icons within the plots show location of sampling site: squares (control sites), circles (150 m from pen), triangles (50 m from pen), crosses (pen).

The bacterial Metabarcoding Biotic Index (b-MBI) and eukaryotic Metabarcoding Biotic Index (e-MBI) were calculated and multivariate analysis on the weighted abundances of the five eco-groups was undertaken. Multivariate analysis indicated strong and consistent clustering of replicates and samples from different kits for each distance along the gradient especially in for the b-MBI (Fig. 6). PERMANOVA results indicated that there were significant differences for in the b-MBI (F = 13.22; p < 0.001) and e-MBI (F = 4.54; p = 0.002) with kit. However, pairwise comparisons showed no significant differences. For distance there were significant differences in both the b-MBI (F = 421.08; p < 0.001) and e-MBI (F = 64.33; p < 0.001) with all distances being significantly different.

Figure 6 Non metric multi-dimensional scaling plots of the bacterial and eukaryotic metabarcoding biotic index.

Non metric multi-dimensional scaling plots depicting (A) bacterial metabarcoding biotic index (b-MBI) and (B) eukaryotic metabarcoding biotic index (e-MBI) (Bray Curtis distance matrix of square root transformation of weights for each eco-group) across a distance gradient at Otanerau fish farm in New Zealand (2015). Q.PS, Qiagen Dneasy PowerSoil Kit (0.25 g); Q.PS.Pro, Qiagen Dneasy PowerSoil Pro Kit (0.25 g); QIA2, Qiagen Rneasy PowerSoil Total RNA/DNA extraction/elution Kit (2 g); FAV2, Favorgen FavorPrep Soil DNA Isolation Midi Kit (2 g); FAV5, Favorgen FavorPrep Soil DNA Isolation Midi Kit (5 g).

Discussion

For molecular methods to be used reliably in monitoring potential degradation of benthic habitats, samples should be representative of the targeted community and the DNA sufficiently pure to ensure that inhibitors do not affect the analysis (McKee, Spear & Pierson, 2015). The use of DNA extraction kits is highly desirable as it standardizes this process and, in some instances, allows the automatization of extraction using robotics. The aim of the study was to explore how the application of four different extraction kits (all commonly used in soil/sediment studies) impacted the composition and structure of prokaryotic and eukaryotic communities in marine surface sediments derived via metabarcoding. In addition, with each kit allowing different amounts of starting material, three different weights of sediment were used (two different weights were tested for the Favorgen kit).

We used benthic sediment samples along an enrichment gradient associated with a salmon farm as a study case, and investigated the quantity and quality of DNA extracted using five different extraction kits. In general, the three kits using higher volumes of sediment (QIA2, FAV2 and FAV5) retrieved higher concentrations of eDNA. However, this higher quantity of DNA was often offset with a lower overall quality, with the Q.PS and Q.PS.Pro kits having the largest number of samples that passed the automatic quality control on the nanophotometer. The QIA2 and FAV2 kits had comparatively low A260/A230, indicating potential contaminants such as humic compounds which absorb in the A230 spectrum (Yeates et al., 1998). The presence of humic acids in sediment samples is a known concern as it can complex with DNA (Lakay, Botha & Prior, 2007) and interfere with subsequent PCR amplification. DNA from all samples was successfully amplified, although PCRs were noted to be less efficient for the kits using higher volumes, indicating that the potential contaminants did not inhibit PCR reactions completely but further studies would be required to assess what was causing the low A260/230 ratios.

We expected that the higher weights of sediment would result in a higher diversity of ASVs. However, the kits using higher starting weights of sediment (QIA2, FAV2 and FAV5) in general revealed lower prokaryotic diversity than the Q.PS and Q.PS.Pro kits, although the significance of these results varied and, except for FAV2, were not observed for the eukaryotes. The lack of correlation between starting material and diversity has previously been reported by Carrigg et al. (2007). The current study assessed two weights using the same kit for the Favorgen kit only, and found no significant difference in the richness between weights. However, further studies using a variety of kits and more weights would be required to confirm this trend. Inefficiencies in the extraction kits using large sediment weights could explain this observation with the possibility that humic acids and other contaminants are binding to the silica filters and reducing the concentration of DNA bound to the filters (Lloyd, Macgregor & Teske, 2010). If the latter hypothesis is true, then the obvious limitation is that research practitioners aiming to appropriately capture micro-patchiness and/or spatial heterogeneity of biological assemblages (i.e., beta diversity) in marine sediments, will have to adapt their sampling size accordingly. For this reason, previous fish farm studies have advocated for the collection of at least 3–5 independent replicate samples per station (Pawlowski et al., 2014; Lejzerowicz et al., 2015; Pochon et al., 2015). In this respect, the detection of higher diversity in the Q.PS and Q.PS.Pro kits has a further advantage as the small volumes used in the Q.PS and Q.PS.Pro kits mean samples can be processed using automated equipment (e.g., QIAcube, Qiagen). The automated methodologies limit the human involvement (i.e., variability) in the procedure and are thus beneficial for monitoring purposes where replicability is vital.

Only a small percentage of ASVs were shared between all kits in both the prokaryotic and eukaryotic datasets. However, these shared ASVs accounted for a substantial (i.e., >60%) portion of the total number of reads. This suggests that all kits are detecting the core communities and that the main differences in detection are in rare and low abundance ASVs, as has been shown in previous environmental metabarcoding studies (Pedros-Alio, 2006; Lynch & Neufeld, 2015). It should be noted that while PCR and sequencing controls were undertaken to detect potential contamination in those steps, no extraction controls were used in this experiment. Therefore, we cannot exclude the possibility that a small proportion of the shared ASVs recovered here are due to residual contamination from kits or equipment, and extraction controls should be sequenced in future studies to evaluate this possibility. The higher number of unique ASVs in the Q.PS and Q.PS.Pro kits could suggest that these kits are able to retrieve a larger number of rare ASVs. However, this may also, in part, be due to stochastic differences in eDNA distribution within sediments rather than extraction differences. The rare ASVs could possibly be found with all kits if increased sequencing depth or further replication (either extraction or PCR replicates) were undertaken. Previous research has shown that increasing replication can give a more reliable estimation of diversity (Lanzén et al., 2017). Kits that require fewer replicates to retrieve a similar diversity are likely to be more cost efficient and thus more suitable for high throughput monitoring applications. These data indicate that the Q.PS and Q.PS.Pro kits provide the best estimation of prokaryotic and eukaryotic community diversity.

Multivariate analysis indicated that while there was a significant interaction between kit and distance from the pens, all the kits showed a similar pattern with different communities along the transect. However pairwise comparisons indicated that the Q.PS and Q.PS.Pro kits had more significant differences amongst distances. Despite a similar trend amongst the kits, there were distinct differences in the relative abundance of taxa. This suggests that there are likely to be taxon-specific variations in cell lysis between the kits, especially in the eukaryotic dataset. This finding is in agreement with other studies that have found a similar trend (Carrigg et al., 2007; Lekang, Thompson & Troedsson, 2015; Ramírez, Graham & D’Hondt, 2018). This could be further tested by using positive extraction controls such as a known mock prokaryotic community, allowing for the assessment of lysis efficiencies amongst kits (Hermans, Buckley & Lear, 2018). In terms of marine monitoring of prokaryotes, the differential lysis of particular groups could affect the classification of samples if taxonomic approaches such as the microgAMBI or Indicator Values (IndVal) are to be used (Aylagas et al., 2017; Borja, 2018). For example, in this dataset Flavobacteriaceae have higher relative abundances in the QIA2, FAV2 and FAV5 kits. This family is classified as tolerant to pollution in the microgAMBI and substantial differences in the relative abundances between kits may impact the conclusions from taxonomy-based approaches. Differences in taxonomy could also further impact conclusions that are based on inferring function based on taxonomic composition using molecular approaches such as Paprica (Bowman & Ducklow, 2015; Laroche et al., 2018). For the eukaryotic dataset the DNA sample will combine a complex mix of extracellular DNA released by macrofaunal organisms, DNA from living organisms (ranging from microeukaryotes up to meiofauna and larger depending on sample size), and fragments of dead organisms. Interestingly, Conoidasida, Novel Apicomplexa Class I and Syndinales are parasitic taxa of invertebrate and vertebrate macro-organism. In the case of Conoidasida and Syndiniales these taxa were more abundant in the kits using larger weights of sediment. This may be due to the fact that these larger weights of sediment would have increased probabilities to sample specimens or fragments of these larger organisms which occur at lower densities in the sediment.

More recently, de novo approaches such as b-MBI and e-MBI which work at the ASV level have been developed to assess marine ecosystem health without the restraint of relying on taxonomic classifications (Keeley, Wood & Pochon, 2018). Multivariate analysis of these de novo approaches indicated that there was a significant difference in the proportion of ASVs assigned to each eco-group. However, pairwise comparisons showed that there was no significant pairwise comparisons amongst the kits and, in similarity with the community composition, the organic enrichment gradient observed with distance from the farm was a stronger determinant of the resulting assessment of health. This suggests that the type of kit used will have limited impact on the management decisions obtained across defined ecological gradients.

Conclusions

Extracted DNA from commercial kits should be of high quantity and provide a repeatable representation of the community in a sample. In this study, we showed that all investigated kits showed a similar pattern of community change along the disturbance gradient away from the fish farm pens, and that the inferred metabarcoding-based biotic indices were also similar amongst kits. This indicated that the organic enrichment gradient had a higher impact on prokaryotic and eukaryotic composition and biotic indices than any individual extraction kit. Further, only a small percentage of ASVs were shared between all kits in both the prokaryotic and eukaryotic datasets. However, these shared ASVs accounted for a substantial amount of total read number, suggesting that the core communities were captured in the DNA extracted by all kits. Nevertheless, while lower overall quantities of DNA were obtained from the Qiagen Power Soil (Q.PS) and Qiagen Power Soil Pro (Q.PS.Pro) kits, likely due to the lower volume of sediment used, the quality of the extracted DNA was higher. This could lead to less inhibition in the proceeding PCR steps. The Q.PS and Q.PS.Pro also had the highest number of unique ASVs.

In conclusion, we advocate for the use of the Q.PS.Pro kit for sampling prokaryotic and eukaryotic communities in marine benthic environments associated with marine aquaculture. While the Q.PS and Q.PS.Pro kit had similar results, the recent discontinuation of the former kit rules out the use of this kit in the future. We base this conclusion on the higher DNA quality values and richness achieved with this kit.

Supplemental Information

Supplemental Information 1 Sediment properties of the stations sampled for DNA analysis.

Click here for additional data file.

Supplemental Information 2 ASV sequences and taxonomy.

Click here for additional data file.

Supplemental Information 3 Concentration, quality and purity of extracted samples determined using the Implen NP80 Nanophotometer (Implen, Munich, Germany).

Automatic quality control identifies lower quality results (in gray) such as air bubbles, impurities, and/or potential contaminations. Note: this quality control looks for perfect relationships between concentration and purity values, and therefore is extremely stringent and does not mean that the extraction has failed.

Click here for additional data file.

Supplemental Information 4 The number of amplicon sequence variants found in each DNA extraction kit for each class of prokaryotes.

Click here for additional data file.

Supplemental Information 5 The number of amplicon sequence variants found in each DNA extraction kit for each class of prokaryotes.

Click here for additional data file.

Supplemental Information 6 The number of amplicon sequence variants found in each DNA extraction kit for each class of eukaryotes.

Click here for additional data file.

Supplemental Information 7 Permanova results on the composition of the prokaryotic and eukaryotic communities with pairwise comparison results.

Click here for additional data file.

Supplemental Information 8 Prokaryotic and eukaryotic rarefaction curves.

(A) prokaryotes and (B) eukaryotes colored by the different extraction kits. Dotted vertical line represents the level of rarefaction undertaken. Q.PS = Qiagen DNeasy PowerSoil Kit (0.25 g), Q.PS.Pro = Qiagen DNeasy PowerSoil Pro Kit (0.25 g), QIA2 = Qiagen RNeasy PowerSoil Total RNA/DNA extraction/elution Kit (2 g), FAV2 Favorgen FavorPrep Soil DNA Isolation Mini Kit (2 g), FAV5 = Favorgen FavorPrep Soil DNA Isolation Mini Kit (5 g).

Click here for additional data file.

Supplemental Information 9 Shannon diversity per kit for the eukaryotic 18S rRNA (Eukaryotes; A) gene and prokaryotic 16S rRNA (Prokaryotes; B) gene.

The symbols designate the distance from the aquaculture pen. Q.PS = Qiagen DNeasy PowerSoil Kit (0.25 g), Q.PS.Pro = Qiagen DNeasy PowerSoil Pro Kit (0.25 g), QIA2 = Qiagen RNeasy PowerSoil Total RNA/DNA extraction/elution Kit (2 g), FAV2 Favorgen FavorPrep Soil DNA Isolation Mini Kit (2 g), FAV5 = Favorgen FavorPrep Soil DNA Isolation Mini Kit (5 g).

Click here for additional data file.

Supplemental Information 10 Community composition at the class level with distance from the fish farm.

(A) prokaryotes, and (B) eukaryotes across the distance gradient. ‘Other’ accounts for those reads that belonged to Classes which were not in the most abundant 10 Classes. Q.PS = Qiagen DNeasy PowerSoil Kit (0.25 g), Q.PS.Pro = Qiagen DNeasy PowerSoil Pro Kit (0.25 g), QIA2 = Qiagen RNeasy PowerSoil Total RNA/DNA extraction/elution Kit (2 g), FAV2 Favorgen FavorPrep Soil DNA Isolation Mini Kit (2 g), FAV5 = Favorgen FavorPrep Soil DNA Isolation Mini Kit (5 g).

Click here for additional data file.

We thank New Zealand King Salmon Co. Limited, for access to their farms, samples and in-kind support. We are also grateful to Olivia Johnson, Deanna Elvines, Lauren Fletcher, Holly Bennett, and Emily McGrath (Cawthron) for their help in collecting the molecular samples. We would also like to thank the editor Dr. Owen Wangensteen as well as the reviewer Dr. Reindert Nijland and another anonymous reviewer for their time and consideration in improving this manuscript.

Additional Information and Declarations

Competing Interests

Author Contributions

Field Study Permissions

DNA Deposition

Data Availability

Xavier Pochon is an Academic Editor for PeerJ. John K Pearman, Susanna A Wood, Anastasija Zaiko, Georgia Thomson-Laing, Laura Biessy, Javier Atalah, and Xavier Pochon are employed by Cawthron Institute.

John K. Pearman analyzed the data, prepared figures and/or tables, authored or reviewed drafts of the paper, and approved the final draft.

Nigel B. Keeley conceived and designed the experiments, authored or reviewed drafts of the paper, and approved the final draft.

Susanna A. Wood conceived and designed the experiments, authored or reviewed drafts of the paper, and approved the final draft.

Olivier Laroche analyzed the data, authored or reviewed drafts of the paper, and approved the final draft.

Anastasija Zaiko conceived and designed the experiments, authored or reviewed drafts of the paper, and approved the final draft.

Georgia Thomson-Laing performed the experiments, authored or reviewed drafts of the paper, and approved the final draft.

Laura Biessy performed the experiments, authored or reviewed drafts of the paper, and approved the final draft.

Javier Atalah conceived and designed the experiments, authored or reviewed drafts of the paper, and approved the final draft.

Xavier Pochon conceived and designed the experiments, prepared figures and/or tables, authored or reviewed drafts of the paper, and approved the final draft.

The following information was supplied relating to field study approvals (i.e., approving body and any reference numbers):

Field experiments were undertaken with the approval of New Zealand King Salmon Co. Limited. Sampling was part of a regular compliance monitoring program for New Zealand King Salmon (NZKS).

The following information was supplied regarding the deposition of DNA sequences:

Raw sequence reads are available at NCBI BioProject: PRJNA657189.

The following information was supplied regarding data availability:

The code used to analyze the fish farm data is available at GitHub: https://github.com/jkpearmanbioinf/FishFarmAnalysis.

The bioinformatics script used for demultiplexing 16S and 18S rRNA paired-end raw sequences, and for quality filtering, denoising and chimera filtering using DADA2 implemented in Qiime2 is available at GitHub: https://github.com/olar785/Optimizing-DNA-extraction-methods-for-assessing-organic-enrichment-in-marine-farm-sediments.

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
