# Peer review of "Comparing sediment DNA extraction methods for assessing organic enrichment associated with marine aquaculture"

_PeerJ, doi:10.7717/peerj.10231_

## Round 0.1 · original submission · Minor Revisions

Both reviewers have provided a list of minor revisions that would need to be addressed, point by point.

I agree with the suggestion that the manuscript would benefit from adding a paragraph to explain the finding of a big fraction of metabarcoding reads coming from metazoans and their parasites, as pointed out by the reviewers. I think that this discussion could be framed out by explaining that what we call "environmental DNA" from the sediments is actually a complex mix of trace (extra-organismal) DNA (released by macrofaunal organisms), plus a big component of community (genomic) DNA from living organisms (including bacteria and microeukaryotes, but also meiofaunal and even bigger organisms). I think that these sentences could help readers to understand a lot of things that are going on during the DNA extraction and PCR amplification of sediment DNA.

I am looking forward to the revised version of this interesting manuscript.

·

Basic reporting

With this manuscript, Pearman et al. are presenting their very interesting results on studying biodiversity in marine sediment, comparing a relevant selection of different DNA isolation kits along a sampling gradient. The manuscript is well written with a clear structure and sufficient context and references provided. The scripts and data analysis procedures are clearly described and/or made available. I was unfortunately unable to access the PRJNA657189 dataset at NCBI SRA, I assume this has not yet been made publicly available?

Experimental design

The research question is well defined, relevant & meaningful. The methods followed are mostly described with sufficient detail. Some unclarities are still present. According to the manufacturer, the Favorgen mini kit that is said to have been used is only suitable for up to 0,5g sediment. As the authors have tested this kit with 2g and 5g, I assume they instead have used the midi version of this kit? Please correct throughout the manuscript and supplementary data.
The other quibble here is that, although the volume of the used PCR-primers is mentioned, the concentration is not stated. Please add the used primer concentration.
I have several points regarding the experimental design. These might be hard to correct now, but would be something to take in to consideration in future experiments. If the authors agree with the points below, it would also be beneficial to add these suggestions to the discussion as to guide future research from anybody reading this manuscript.
The DNA concentration and quality directly after the isolation was measured with a nanophotometer. For accurate DNA concentration using a fluorometric approach such as a Qubit (used elsewhere in the experiments) would be much more accurate. Although initial differences in DNA concentration are unlikely to have a major effect on the observed diversity after metabarcode amplification, measurement of these metrics should still be performed using the most appropriate method available. It will also provide additional information on the DNA quality and contamination.
Another improvement could be the use of a isolation controls. This will provide valuable insights in the DNA contamination present in the sampling equipment and the so called kit-contamination. Especially when comparing different kits this would have been interesting. Especially in experiments using eDNA (with usually very low concentrations of target DNA) this would be very valuable.
Finally, for the mentioned difference/bias in extraction efficiency for different species (l.418-420), an inclusion of extraction controls using e.g. zymo microbial standards would have provided valuable information about this bias for the DNA isolation kits used.

Validity of the findings

The findings described in this manuscript are well described and all statistical methods used described in clear detail. Some processed data is provided in the supplemental materials. A table with the sequences of all identified ASV’s is not given, and would be an interesting addition to the supplemental data. Also, as mentioned above, the raw short read data could not be retrieved from NCBI.
The conclusions are clear and based on the data obtained in this study. I would say that even more emphasis could be placed on the fact could that all kits are able to show similar core communities and as such, also the gradient present is at the research site is resolved regardless of the kit used.
It is mentioned that less pure DNA could lead to effects such as PCR inhibition. As the Favorgen kit was clearly producing less pure DNA (line 372-378, Table S2), was such an effect also observed in the PCR results? E.g. in less efficient amplification/lower overall yield/smearing or less pronounced bands after PCR? It would be interesting to report on this if such effects were observed.

Additional comments

From the abundance plot (fig 4) for the eukaryotes, it is apparent that the classes listed in the top 10 most abundant reads only represent very small organisms. To me, it seems highly likely these small organisms are present on and inside the sampled sediment and mostly were alive and intact during sampling, just like the bacteria that were identified. As such, I would not consider this an eDNA study, and would describe this as a study of microbiome and bulk benthos. For an eDNA sample, especially taken at a fully stocked fish pen, I would at least expect high numbers of reads for the animalia, more specifically the exact fish species present in the pen. I am curious is fish DNA was found at all in this study.

Although I understand this paper is mainly about the difference between the DNA isolation kits, the obtained results related to the Enrichment State index and the findings in a ecological context are also of relevance. It is clear from e.g. fig. 2 that the CTL site has a much higher biodiversity compared to the site directly near the fish pen. I would suggest to either shortly discuss these results in the manuscript, or, alternatively, mention more clearly why these are deliberately not discussed.
Also, the use of bacteria at class level in my opinion is unlikely to give a real meaningful picture of the ecology, as there is a very high functional diversity of bacterial strains inside the observed classes. Clearly using bacterial diversity is able to show a gradient, and this might be sufficient in the context of the ES index. It would be worthwhile to shortly discuss this issue in the manuscript to clarify the intention of measuring bacterial diversity.

In the abstract, line 45-46, “to” should be removed.

Reviewer 2 ·

Basic reporting

no comment

Experimental design

no comment

Validity of the findings

no comment

Additional comments

In this paper, the authors present the comparison of different sediment DNA extraction methods evaluating their efficiency in the context of using sediment DNA metabarcoding to assess the organic enrichment associated with salmon farms activities. The topic is timely and of great importance for future developments of DNA-based biomonitoring. The reported value of 3% of shared ASVs between the kits shows to which extend this aspect is important for any kind of comparative studies.
The paper is well written and, in my opinion, does not contain any technical or analytical flaws. However, there are few important points that should be taken in consideration while preparing the revised version:

1) The title does not really reflect the content of the paper. The authors do not optimize any DNA extraction method but rather compare the existing protocols. Moreover, the wording “marine farm sediments” sounds very unclear. I would suggest something like.
“Comparing sediment DNA extraction methods for assessing organic enrichment associated with marine aquaculture”

2) The authors should carefully revise the statements concerning the composition of eukaryotic assemblage obtained from sediment DNA. In fact, the figure 4 shows that among the 10 most abundant classes within eukaryotes, there are several metazoan groups (nematodes, copepods, polychaetes). There are certainly not the micro-eukaryotes and their presence in eDNA dataset can be affected by the volume of extracted sediment.
By the way, the apicomplexans that also very abundant in the sediment DNA data are mainly parasites of invertebrates. So, the eukaryotic DNA in marine sediments is dominated by metazoans and their parasites. This aspect might be worth discussing in the paper.

3) The impact of the weight of the sediment on DNA diversity is based on comparison of different type and brand of kits. These results need to be confirmed by the analysis of different sediment weights using the same kit or at least the same brand of kit. Those that have done these analyses with Qiagen PowerMax Soil kit will probably disagree with your conclusions.

4) As far as I understood, all kits provided DNA that could be amplified during this study. So, the discussion about the low quality of DNA generated by some kits (FAV 2 and FAV5) and potential inhibition should be moderated as long as there is no strong evidence for such inhibition. Otherwise, the producers of these kits could feel unfairly treated.

5) The authors omit at least two important papers about DNA extraction from marine sediment. These papers have to be referenced and their results discussed in the paper:
• Lekang et al. Aquatic Microbial Ecology 2015
• Hermans et al. Mol Ecol Res 2018
The authors could also mention the papers that focus on ancient sediment DNA, which present several extraction protocols that can be used for recent and fossil sediments:
• Epp et al. Meth Mol Biol 2019
• Haile, Meth Mol Biol 2012
• Armbrecht et al. Mol Ecol Res 2020


Minor points:
Line 26 – “selection of sediment extraction weight and the DNA extraction method” sounds somehow weird – could be replaced by “selection of sediment DNA extraction method” (which anyway contains sediment weight)
Line 45 – “with the to distance” – something is lacking here
Line 359 – benthic health?
Line 420 – taxon-specific
Line 435 – the factor kit ?

---

## Round 0.2 · accepted · Accept

All changes recommended by the reviewers have been addressed satisfactorily. Congratulations.